# Structural Features of Fatigue Crack Propagation of a Forging Die Made of Chromium–Molybdenum–Vanadium Tool Steel on Its Durability

**DOI:** 10.3390/ma16124223

**Published:** 2023-06-07

**Authors:** Marek Hawryluk, Marzena Lachowicz, Aneta Łukaszek-Sołek, Łukasz Lisiecki, Grzegorz Ficak, Piotr Cygan

**Affiliations:** 1Department of Metal Forming, Welding and Metrology, Wroclaw University of Science and Technology, Lukasiewicza Street 5, 50-370 Wroclaw, Poland; marzena.lachowicz@pwr.edu.pl; 2Faculty of Metals Engineering and Industrial Computer Science, AGH University of Science and Technology, Av. Mickiewicza 30, 30-059 Krakow, Poland; alukasze@metal.agh.edu.pl (A.Ł.-S.); lisiecki@agh.edu.pl (Ł.L.); 3GK FORGE Sp. z o.o, Przemysłowa 10 Street, 43-440 Goleszów, Poland; gficak@gkforge.pl; 4Kuźnia Łabędy S.A., Mechaników 9 Street, 44-109 Gliwice, Poland; p.cygan@kuznialabedy.pl

**Keywords:** durability of a forging die, hot work steel, forging in a double system, wear, fracture, destructive mechanisms

## Abstract

The paper presents the results of tests on a die insert made of non-standardised chrome-molybdenum–vanadium tool steel used during pre-forging, the life of which was 6000 forgings, while the average life for such tools is 8000 forgings. It was withdrawn from production due to intensive wear and premature breakage. In order to determine the causes of increased tool wear, a comprehensive analysis was carried out, including 3D scanning of the working surface; numerical simulations, with particular emphasis on cracking (according to the C-L criterion); and fractographic and microstructural tests. The results of numerical modelling in conjunction with the obtained results of structural tests allowed us to determine the causes of cracks in the working area of the die, which were caused by high cyclical thermal and mechanical loads and abrasive wear due to intensive flow of the forging material. It was found that the resulting fracture initiated as a multi-centric fatigue fracture continued to develop as a multifaceted brittle fracture with numerous secondary faults. Microscopic examinations allowed us to evaluate the wear mechanisms of the insert, which included plastic deformation and abrasive wear, as well as thermo-mechanical fatigue. As part of the work carried out, directions for further research were also proposed to improve the durability of the tested tool. In addition, the observed high tendency to cracking of the tool material used, based on impact tests and determination of the *K*_1C_ fracture toughness factor, led to the proposal of an alternative material characterised by higher impact strength.

## 1. Introduction

At present, with the observed development of forging and in view of the high prices of energy and raw materials, more and more often, for the production of responsible machine elements for many industrial branches, die forging at elevated temperatures is applied [1]. The competitiveness of the forging processes is also dictated by the high utilitarian properties of the obtained forgings in respect of other manufacturing technologies [2]. This said, also in the case of forging processes, there is a search of new solutions connected with reducing the energy consumption and protecting the environment, which translates to, e.g., a reduction in the amount of the charge material or application of the optimal number of operations in respect of the durability of the forging tools [3]. That last aspect especially, i.e., the time period of tool operation, constitutes about 20 to 40% of the process costs [4]. In extreme cases, when the tools are exceptionally loaded, e.g., during the forging of elements with complex shapes, or in test series, this cost can increase to as much as 80% [5,6]. Of course, the costs of the whole process directly translate to the unit prices of the forgings. For this reason, the crucial factors are the forging tool durability and a properly designed forging technology.

In the hot die forging processes the punches and dies as well as other elements of instrumentation very often work under extreme conditions, as they are exposed to cyclic high mechanical loads (sometimes exceeding 2200 MPa) and thermal loads in the range of 60–100 °C to 1300 °C [7,8]. In the analysis of a hot forging process in its technological aspect, one should point out that the charge is heated much above the recrystallization temperature (often over 1000 °C), which, after it is placed into the tool and the deformation begins, causes the temperature on the tool surface to intensively increase [9]. This creates serious conditions of thermo-mechanical load and consequently microstructural degradation of the die surface, causing significant damage of the operational surface through abrasive wear, plastic deformation, and thermo-mechanical fatigue [10]. Therefore, the tools are exposed to the operation of many destructive factors, which cause their wear, both premature and after a longer operation time. The areas especially subjected to wear are the working surface and the surface area of the tool, and so, most of the mentioned destructive mechanisms refer to those areas of the tool [11,12]. Exploitation is an indispensable phenomenon accompanying the production of products and is most often associated with the maximum use of a machine/system/tool in a specific time, after which it ends with their partial or complete wear. Therefore, in the technical literature, a lot of space is devoted to the issue of exploitation and a number of studies have been carried out on the determination of significant parameters affecting this phenomenon, as well as industrial research and development works that allow researchers to increase the exploitation time [11,12,13]. The most commonly occurring and main destructive mechanisms include abrasive wear [14] and plastic deformation, during warm forging [15,16,17], as well as plastic deformations in the case of hot forging [18,19]. Other dominating destructive mechanisms are thermo-fatigue cracks [20,21,22] and thermo-mechanical fatigue [23,24]. Among them, the most thoroughly analysed is the wear mechanism, which occurs mostly in the cold forging processes [25,26], but also in the hot forging processes, although it is not a dominating destructive mechanism [27]. Abrasive wear is often viewed as the key destructive mechanism, as it is relatively easy to measure; however, the most important factor, often deciding about the tool’s removal and thus interruption in the production process, is thermo-mechanical fatigue. The literature provides many studies referring to a complex analysis of the main destructive mechanisms of the tool in hot die forging processes [28,29,30]. Performing numerous studies and conducting a complex and in-depth analysis of the whole forging process, especially of the forging tool durability, is crucial to obtain an answer and undertake appropriate measures. In order to avoid these phenomena, various methods and techniques are applied, which consist of introducing new solutions into the process itself, ensuring control and stability, or producing and applying appropriate protective coatings, as well as applying alternative materials. These methods aim at increasing the durability of forging instrumentation and protect it from destructive mechanisms, thus guaranteeing the proper quality of the forgings [31,32,33,34].

Therefore, the activities and studies performed in order to improve tool durability are concentrated mostly on optimising the forging technology, including the selection of the optimal tool material (in respect of the properties and the cost), its thermal and thermo-chemical treatment, as well as protection of the surface layer of the tools, which is especially exposed to the operation of destructive factors [35,36,37,38]. The operation conditions of forging tools as well as the other instrumentation require that the technologists and constructors select a tool material which will meet the expectation connected with their hard work, that is high resistance to cyclic high mechanical and thermal loads, and which will minimise the effect of the operation of destructive mechanisms [4,39,40]. At present, the most commonly used material grades for forging punches are 1.2343, 1.2344, 1.2365, 1.2367, 1.2999, etc., which are characterised by very good mechanical properties (high tensile strength and hardness, high abrasion resistance, and high yield strength equalling over 2000 MPa). The fulfilment of these requirements is realised through the selection of an adequate thermal treatment (high quenching and two-fold tempering) and also through an appropriate content of alloy elements for the tool material [41,42], as well as through the elaboration of a proper and adequate forging technology, as often, an insufficient durability or premature wear of the tools is often caused by a poorly followed technology or the human factor [43]. In turn, the protection of the surface layer of the tools usually takes place through the use of surface engineering techniques, which include nitriding [44] and hybrid layers [45]. Nitriding is a relatively known and effective thermo-mechanical treatment technology. We can find many studies referring to increasing the operation time with the use of a nitrided layer [46,47,48]. In turn, hybrid layers are a much younger method enabling a durability increase. Usually, they combine a nitrided layer with other, much thinner coatings, often based on chromium or boron. Additionally, in this case, we can find many works pointing to an effective operation and protection against one or several simultaneous destructive mechanisms [49,50,51,52,53].

The presented analyses show that it is advisable to perform further advanced studies and research and development investigations in this area, which will contribute to an even better understanding of the phenomena taking pace in tool steels for hot operations assigned for forging tools during their work as well as the effect of the conditions present during the forging processes on the changes in the surface layer and the tool microstructure [43,54]. This should also make it possible to develop and select optimal solutions in respect of the tool material, which will characterise in the mentioned properties ensuring high resistance to destructive mechanisms and increasing the durability of forging instrumentation.

The aim of the work is a comprehensive analysis of premature wear and cracking of the die insert used in the process of multiple hot forging, in a dual system, of elements of motor trucks on a mechanical press. 

## 2. Materials and Methods

The test material is a die insert used for preliminary forging made of hot working tool steel. The die insert was used in the process of multiple hot forging, in a dual system, of elements of motor trucks on a mechanical press LZK 1000 with the nominal forming force of 1000 t. The die insert is mounted in a specially designed casing (Figure 1a).

The temperature scope of the insert equalled about 200–250 °C, which introduced the necessity of its pre-heating before the beginning of the production process in order to eliminate the risk of brittle cracking during the process start-up. Due to the complexity of the multiple forging process and the possibility of the risk of deformation of the die inserts as well as the shape of the forging (relatively complicated, compact, with changeable sections), it was important to properly design the preliminary pass. Figure 1b shows the insert after the operation time, i.e., after it was removed from the production process due to its wear and cracking, which constituted the cause of the tool’s damage. The figure marks the areas of further macro- and micro-observation. The insert characterises in a significant increase in the rounding radii and the depth, as well as decreased dimensions of the impression’s width in respect of the die (finishing) insert and an appropriately designed inside groove for the flash. The durability of a worn die inserts equalled 6000–7000 forgings (a dual system—12–14 thousand single forgings). In industrial practice, the wear of preliminary roughing passes determines the changeovers of the press and thus the downtimes in the production process. During the forging, the highest loads of the device are also present. Based on the preliminary macroscopic tests, for a detailed analysis, those areas of the die were selected which reflected the characteristic traits of wear for all the areas for which operation surface state evaluations were performed (samples: 2a, 5b). For the fractographic analysis, those samples were chosen for which cracking had been initiated (samples: 3a, 3b)—Figure 1b. Within the realised investigations, the following research methods were applied:Macroscopic analyses with a measurement of the wear degree/material loss on the working surface of the tool by means of the 3D scanning method with the use of a measuring arm ROMER Absolute ARM 7520si integrated with an RS3 scanner and a comparison of the geometry of the scan with the CAD model;Numerical simulations with the use of the QForm program with a special consideration of the temperature distributions and the criterion of cracking of the die insert material (together with the modification of the subroutine consisting in adding the elastic part of the deformation to the cracking model according to Cockcroft–Latham criterion (C-L))An analysis of the chemical composition conducted with the use of an analyser (glow discharge spectrometer) GDS 900 by LECO;Observations of the tool surface state, as well as fractographic tests performed by means of a stereoscopic microscope Leica M205 C and a scanning electron microscope ThermoFisher Phenom XL;Microstructural observations with the use of a light microscope Leica DM6000M. To that end, the die insert was incised along the shorter side to prepare samples for the tests. The grinding and polishing, in order to obtain traditional micro-sections, was conducted on a grinder–polisher Struers 350. For the etching, a picric acid solution was used;Hardness measurements made by means of a hardness tester LECO LC100;The impact test was carried out in accordance with PN-EN ISO 148-1:2017-02 Impact using the RKP 300 Charpy hammer and determination of the fracture toughness factor *K*_1C_.

## 3. Experimental

### 3.1. Tool Material Characteristics

In terms of the chemical composition, we cannot ascribe the examined die to a normalised steel grade. The most similar grade in respect of the chemical composition is steel 1.2367 (X38CrMoV5-3) according to PN-EN ISO 4957:2018-09 [55]. However, this grade demonstrates a slightly higher value of silicon content and a higher content of molybdenum compared to the tested steel. The remaining elements exhibit an agreement with this steel. Other normalised tool steel grades Cr-Mo-V mentioned in Table 1 characterise in a higher carbon content. Both 1.2343 and 1.2344 steels demonstrate a much lower content of molybdenum compared to the examined steel and steel 1.2344—also a higher content of vanadium. These grades also exhibit a higher content of silicon. The best agreement is demonstrated by the tested material in the case of the chemical composition declared by the producer of Unimax steel. In respect of the microstructure, the tested material characterises in a tempered martensite microstructure (Figure 2). The hardness of the examined material equalled 600 HV1, which corresponds to the hardness of about 54 HRC. The high content of molybdenum favours the formation of carbides with high stability.

### 3.2. Macroanalysis by 3D Scanning and Surface Layer Morphology

In order to determine and verify the durability of the tested tool—a die insert working in a dual system—a macro-observation and an analysis of the geometrical changes were performed by way of scanning of the working surfaces. The 3D scanning results have been presented in Figure 3. The highest wear attesting to a big material loss occurs in the central part between the blanks and locally exceeds even almost 4 mm. This wear is a result of the intensive flow of the charge material during upsetting. In the scan image, it is also possible to see a crack, which appeared in the area of the smallest section of the tool, most probably as a result of cyclic high temperature gradients as well as high pressures.

Additionally, the microstructure of the subsurface layer was examined in the areas marked in Figure 3.

Other areas in which we can observe large material losses are also places where the forging material flows over the bridge into the flash groove. In these areas, typical abrasive wear dominates, and the material loss in the nominal direction in within the scope of 1.4 to 1.8 mm.

### 3.3. Numerical Modelling of the Occurrence of Cracks during Forging

Cracking is a very complex phenomenon in metal plastic forming processes. The prediction of cracks in materials in numerical simulations requires the consideration of many phenomena and conditions determining the exceeding of the critical value of deformation, which leads to material separation. The fracture criteria are used for the verification of areas with an increased fracture risk [56]. These criteria take into consideration both the microstructural phenomena (e.g., the size of internal voids) and the unique state of stresses and deformations favouring material separation [57]. Cracks in the working area of the tools in metal plastic forming processes are common.

Predicting the areas with an increased risk of damage makes it possible to modify the technology already at the designing stage. In the presented studies, the mechanisms of the formation of cracks in the working area of a forging die were examined (Figure 4a). Based on the numerical simulation of the process performed in the QForm program, three areas in the crack line were selected. Areas A and B are connected with the displacement of the tool material. In area C, the highest value of the crack criterion was observed (Figure 4b). The literature provides many criteria of cracking. The study proposes the application of a modified Cockcroft–Latham criterion (C-L) for the prediction of the risk of tool damage during forging. The C-L criterion is considered as the ductile cracking criterion for the prediction of material cracking during plastic deformation [58,59]. The criterion assumes that the highest risk of material damage occurs in the areas of principal stress concentration σ_1. In these areas, intensive local material elongation is expected. The C-L criterion is successfully used to predict cracking in metal plastic forming processes. The criterion assumes that the greatest risk of material damage occurs in the areas of $\sigma_{1}$ principal stress concentration. Intensive local stretching of the material is expected in these areas.
(1)$$\int_{0}^{{\overset{-}{\varepsilon }}_{f}}\sigma_{1}d\overset{-}{\varepsilon }=C_{1}$$

The C-L criterion is successfully used to predict fracture in metal forming processes. It is assumed that the material is going to crack when the critical plastic strain ${\overset{-}{\varepsilon }}_{f}$ is exceeded. The performed numerical analyses showed that in the area of fracture, the material of the tools deforms both elastically and plastically. Figure 5 shows an elastic strain and $\sigma_{1}$ principal stress distribution on the surface of tool during forging. The proposed modification of the C-L criterion assumes the inclusion of both elastic and plastic strain components in the criterion.
(2)$${\overset{-}{\varepsilon }}_{f}={\overset{-}{\varepsilon }}_{e}+{\overset{-}{\varepsilon }}_{p}$$

The plastic deformation of the tool material takes place in the subsurface layers under the effect of high local pressures. In the same way, the modified criterion will predict the risk of tool cracking even when, during the process, plastic deformation of the tools does not take place.

The boundary conditions for the simulation of the forging were selected based on the analysis of the industrial process. The modified X37CrMoV51 (1.2343) steel model was adopted as the material of the tools. The first mechanism of fracture was dependent on the displacement of the tool material connected with elastic deformation. As shown in Figure 6b,c, deformation of the tool was observed in different directions. From the point of view of the fracture, the most important is the boundary line between the positive and negative values of displacement in the analysed direction (Figure 6a). Figure 6b presents the displacement distribution in X direction. The boundary between the positive and negative values of displacement is located in the fracture line in Area A (Figure 6b).

A similar situation has been presented in Figure 6c. In this case, a displacement in the Y direction can be observed. The boundary is strongly connected with Area B of the fracture line. It can be assumed that, based on the location of the boundary between the positive and negative values of the tool material displacement, the direction and location of the fracture can be predicted. It is important that Area A and Area B in the fracture line are located outside the die cavity. In this area, elastic strain of the tool material is very low. In the second fracture mechanism, the presented assumptions made it possible to adopt the C-L criterion to determine the risk of fracture in the area of the tools during forging. The place where the fracture occurs (Area C—Figure 4b) in the die has been identified as the area of increased risk of damage (Figure 5d). The highest value of the fracture criterion is in the die cavity. In this area, a concentration of elastic strain and principal stresses are also observed (Figure 4). Considering these results, we can suppose that the fracture began in Area C and propagated to Area A and B because of the local displacement of the material. Based on the above analysis, it can be concluded that the prediction of the risk of fracture of forging dies during the process is possible. Thanks to the modification of the fracture criterion, the area of a high risk of fracture indication can be selected. The analysis of the material displacement in different directions shows the lines of the predicted crack propagation. Additionally, the distributions of temperature fields in the tool during forging were analysed (Figure 7).

As we can notice, the temperature changes very dynamically, especially on the bridge, which is caused by the intensive flow of the forging material and its simultaneous displacement into the impression and the flash (Figure 7b). In turn, after filling the impression, the material does not flow so intensively into the flash anymore, hence the lower temperature on the bridges. In should be emphasised that the whole deformation process lasts about 0.15 s in this operation, which, with high value of stresses, can intensify the phenomenon of material cracking in the most loaded area, and this is a verification of the situation occurring in the industrial process.

### 3.4. Surface Change Characteristics

On the basis of the preliminary macroscopic tests, for the analysis of the operation surfaces, sample 5b was selected because of its clear traces of wear and its location in the central part of the tool. The analysed areas have been marked with symbols from S1 to S6. Macroscopically, on the external operation surface, clear surface changes were observed (Figure 8). Both the working impression and the bridge of the insert, most exposed to wear, exhibited on their operation surfaces traces of plastic deformation as well as cracks suggesting the occurrence of thermal fatigue (area S2). On the edge of this area, spallings of the tool surface were observed (area S1)—Figure 9.

On the cross section, they were accompanied by cracks propagating in different directions in respect of the surface (Figure 10). The observations performed with the use of scanning microscopy methods demonstrated that the thermal fatigue was accompanied by the formation of stick-ons of the forged material (Figure 11 and Figure 12).

The presence of these stickers was confirmed by tests using the EDS method (Figure 13). The tool was used in forging elements made of high chromium stainless steel. The dark areas were dominated by elements derived from the tool material (iron, molybdenum, and vanadium). Increased chromium intensity was observed in the sticking areas. In these areas, lower iron intensity was observed due to the higher content of alloying elements in the forged material.

The tests performed on the cross section of the tool showed that the cracks present on the surface are perpendicular to the surface and demonstrate a similar length of about 300 µm (Figure 14 and Figure 15). The forming fatigue cracks began to open yet did not propagate into the material. The subsurface area exhibited also plastic deformation of the surface.

The lower surface of the tool also demonstrated traces of plastic deformation with characteristic wrinkling of the surface, occurring in area S4 (Figure 16).

This led to the formation of characteristic grooves arranged along the direction of the plastic material flow of the surface layer. Directly behind that area, a fatigue crack propagated along the change of the tool profile in area S3. In the FEM analyses, the highest material effort was observed in these areas (Figure 5d). In the remaining area of the lower part of the tool (S4), typical abrasive wear was observed, which led to a significant material loss (Figure 3). We cannot exclude the possibility that these areas also underwent plastic deformation; however, the processes of tribological wear demonstrated a higher intensity in this area. This resulted in the formation of irregularities on the external surface of the tool, which were not accompanied by plastic demonstration. These areas constituted potential notches favouring the initiation of cracks (Figure 17 and Figure 18).

### 3.5. Fractography

The fractographic tests were performed based on the fracture separating element 3a and 3b, as well as the fracture formed on element 2a. The created fracture is multi-plane and multi- origin. Along the edge of the crack’s beginning, numerous ratchet marks occur. The crack initiated as a fatigue crack further propagates as a brittle crack with clear chevron marks, which are formed in the case of plastically formed elements (Figure 19, Figure 20, Figure 21 and Figure 22). The surface of the fatigue area is very narrow, which is connected with the significant overloads exerted onto the tool. Figure 19 shows the direction of crack propagation from the focus across plane I. The origin is constituted by the internal surface of the insert die, from which the crack propagated into the material. At the last stage, a change of its direction took place and plane II was created. The fractures located opposite them (III, IV) have been presented in Figure 19. Area III of the fracture constitutes a counter-surface for area I, whereas the fracture area marked as IV—for area II. The crack was initiated in the area of the section change, constituting zones of local stresses. The crack propagation in the brittle crack in unstable and for this reason, the initiated crack can propagate through the whole tool section even under the effect of internal stresses and even with reduced external stresses. Figure 20 shows the fracture area separating different planes of the forming crack on element 2a. The areas (V–VII) present the fracture zones which were connected with the propagation of a crack from tool surface S5 and the formation of ratchets as a result of a change of the crack trajectory. The change in the direction of the crack propagation should be connected with the change in the section of the examined die occurring in this area. A different area (IX) was formed as a result of the development and propagation of another fatigue crack. The macroscopic observations revealed in this area the presence of certain features typical of ductile fractures. In this area, despite the macroscopically brittle character of the immediate zone, locally, it was possible to observe traces of plastic deformation (Figure 21). The secondary ridges forming the chevron marks are parallel to the direction of crack growth and demonstrate traits of plastic deformation leading to the formation of delamination in the fracture surface. The third zone (VIII) constitutes a crack whose front was ended in areas V–VII. All the independently developing cracks met, forming an expanded surface topography (Figure 22).

The observations performed on micro-sections made on insert die sections showed that the formed fatigue cracks initially developed as single cracks (Figure 23). As a result of fatigue, their consecutive branches appeared, creating clusters of microcracks forming one common crack. This is connected with the fact that the fracture was formed under the conditions of multi-origin state of effort with a simultaneous operation of contact stresses. Such a form of a developing crack favoured the formation of numerous auxiliary faults, which were observed at the stage of macroscopic tests. In consequence, the micro-image often resembles cracks formed during SCC (stress corrosion cracking)—Figure 24 and Figure 25. The microscopic observations confirm that the forming cracks have a transcrystalline character. The crack nucleation was accompanied by the formation of slip bands caused by the phenomenon of material deformation, which is visible in the non-etched state in Figure 26. The formation of slip bands will be accompanied by material strengthening on the crack front.

### 3.6. The Charpy V-Notch Impact Test

The results of the tests and analyses carried out suggested the need to perform additional impact tests for the steel used for the matrix insert, for which a low value of impact strength was observed, which is an indicator of material ductility. Table 2 presents the results of tests at elevated temperatures of 100–300 °C, assuming such values as the typical operating temperature range (heating of tools before forging) of die inserts. The presented test results showed that the material used for the analysed tools did not show an increase in ductility when the temperature was increased. For comparative purposes, impact tests were also carried out for a new material from which new, more fracture-resistant tools would be made in the future. Three samples were tested for each variant.

When exposed to dynamic loads, high stress values combined with areas of high stress concentration appearing in the matrix areas will result in the growth of cracks, which may eventually lead to its destruction, which was the case with the tested matrix. However, if the cracks do not exceed the critical values determined by the limit value of the stress intensity factor *K*_1*c*_, then their presence will not lead to its catastrophic destruction. For this reason, knowing the value of this parameter is important from the point of view of increasing the durability of the matrix. Taking this into account, the results of the obtained impact tests were used to determine the correlation between the impact strength and fracture toughness of both steels. For this purpose, an empirical relationship was used to predict the value of *K*_1*c*._

There are many dependencies used to predict this parameter characterising the fracture toughness on the basis of impact tests [60,61]. However, conducting research on the properties of steel at lower-shelf and transition-temperature is therefore of no importance in the context of hot forming. In addition, some of them determine the *K*_1*c*_ parameter based on the value of Rp_0.2_, the high value of which with low impact toughness typical for tool steels significantly limits the applicability of these methods. Qamar et al. [62] determined an empirical relationship for H13 steel which well reflects the behaviour of hot work tool steels. Therefore, based on the proposed approach presented in [63,64,65], *K*_1*c*_ values were determined using the Formula (3), which are summarised in Table 2.
(3)$$\left(\frac{K_{1c}}{HRC}\right)=2.4\left(\frac{CVN}{HRC}\right)+0.17$$
where: *K*_1*c*_—crack resistance coefficient, *HRC*—hardness, and *CVN*—energy in the impact test (J),

The assessment was based on the assumption that the hardness of the tested steel used for the matrix insert does not change in the analysed temperature range and amounts to 54 *HRC* according to the hardness measurement. Based on the catalogue data of the manufacturer of the proposed new steel, it was determined that it has a similar hardness after heat treatment in similar conditions to the tested steel. Taking this into account, the assumed hardness after tempering of 54 *HRC* and stable in the tested temperature range was also assumed in the calculations.

The test results obtained indicate that the proposed alternative material for tools shows an increased sensitivity to the increase in impact toughness with temperature, as well as a higher fracture toughness factor in relation to the currently used one, which predisposes it to be used as die inserts for the analysed process, especially in the context of increased resistance to cracking, which was the reason for withdrawing the tested tool. At the same time, increased impact strength may be associated with greater susceptibility to plastic deformation at elevated temperatures, in particular in those areas of the surface layer of the tool where the longest contact of the “hot” deformed material of the forging takes place. Therefore, it may be necessary to harden the surface of the tool by thermo-chemical treatment (nitriding). Therefore, the use of steel with higher impact strength will increase the resistance to cracking, and surface hardening will protect the die against plastic deformation, which was observed in the worn surface layer of the die.

## 4. Discussion

The study performs a complex analysis of the lower die insert (II operation) used in the process of hot multiple forging in a dual system of elements for motor trucks on a mechanical press with the nominal forming force of 1000 t. The die insert is mounted in a specially designed casing together with the tools. Before the forging, the die is heated to the working temperature of 200–250 °C and during the forging it is lubricated with a graphite mixture. During the process, the tool is subjected to high cyclic thermal and mechanical loads as well as intensive friction in the surface layer of the impression and on the bridge as a result of the intensive flow of the deformed material. The performed tests and analyses made it possible to conclude that after producing 6000 forged elements (2 forgings) the tool underwent cracking as well as significant degradation of the surface as a result of the occurrence of different destructive mechanisms. On this basis, it was stated that the dominating destructive mechanism is fatigue cracking caused by high material effort as well as high temperature gradients. Another destructive mechanism determining the tool’s removal is abrasive wear in the central part of the die—on the connection of the dual system of the forgings, as well as plastic deformations caused by the long time period of contact of the deformed material in the impression of the lower die insert. The formed fracture is multi-plane and multi-origin, with numerous ratchet marks created as a result of a change in the crack trajectory. The crack, initiated as a fatigue crack, further propagated as a brittle fracture, with clear chevron marks. The surface of the fatigue area was very narrow, which was connected with the significant overloads working on the operating tool. The microscopic observations made on insert die sections showed that the formed fatigue cracks initially developed as single transcrystalline cracks. Next, as a result of the progressing fatigue, their consecutive branches appeared, forming clusters of microcracks, which formed one common crack. This is a consequence of the fact that the fracture was created under the conditions of a multi-axial effort state with a simultaneous operation of contact stresses. Such a form of the created crack favoured the formation of numerous auxiliary faults at the stage of macroscopic tests. Therefore, it is justified to use a new material for tools, which is why the authors proposed an alternative steel—Orvar 2 m.

## 5. Conclusions

The conducted research made it possible to draw the following conclusions:The analysis of the chemical composition showed that the die was made of tool steel for hot working. The most similar grade in respect of the chemical composition is steel 1.2367 (X38CrMoV5-3). However, the examined steel characterised in a lower content of silicon and molybdenum compared to this grade. The material demonstrated the best agreement in terms of the chemical composition with that declared by the producer of Unimax steel. In respect of the microstructure, the tested material characterised in a microstructure of tempered martensite with the hardness of 54 HRC.On the basis of the analysis of the 3D scanning results, a large material loss was stated in the central part between the semi-finished products, which locally equalled even over 2.8 mm. This is most probably abrasive wear being the result of the intensive flow of the charge material in these areas. In turn, the microscopic tests also demonstrated plastic deformations, which affected the changes in the tool geometry. A more thorough analysis of the tool geometry also makes it possible to state that the shape and construction of the tool can have an important effect on the formation of fatigue cracks—the dislocation effect.The numerical simulations performed in the QForm program enabled an examination of the mechanisms of the crack formation in the working area of the forging die. The first crack mechanism was dependent on the tool material’s displacement combined with plastic deformation. The application of a modified Cockcroft–Latham criterion (C-L) provided the possibility to predict the risk of tool damage during forging. The conducted numerical analyses showed that in the fracture area the tool material deforms both elastically and plastically. The simulation results also demonstrated an important effect of the temperature changes both on the possibility of crack formation and local material tempering as a result of plastic deformations.The bridge, most subjected to wear, showed traces of plastic deformation pointing to the occurrence of thermal fatigue, which was accompanied by the formation of stick-ons of the forged material. On its external edge, spallings of the tool surface were observed, which, on the cross section, were accompanied by cracks propagating in different directions in respect of the surface.The lower operation surface of the tool exhibited traces of plastic deformation with the characteristic surface wrinkling. This led to the formation of characteristic grooves arranged in the direction of the flow of the surface layer material. The plastic deformation and the change of the tool profile taking pace directly on it (clearly modified as a result of the formation of grooves in this area) led to the initiation of a fatigue crack in this area. It should be emphasised that, also in the FEM analyses, these areas exhibited the highest material effort.In the remaining tool area, typical abrasive wear was observed. We cannot also exclude the occurrence of plastic deformation in these areas. However, because of the increase in the intensity of the abrasive wear of the surface layer, it was not identified during the realised studies.The conducted microscopic examinations make it possible to state that, despite the macroscopically brittle character of a larger part of the formed fracture, the nucleation of the fatigue cracks was accompanied by the formation of slip bands. It should be expected that this effect is connected with the material deformation, which will result in material strengthening on the front of the developing crack.

## 6. Directions for Further Research

Further studies will be concentrated on the search of methods and ways of increasing the crack resistance and the general tool durability.

-For this reason, it seems that, in the first place, we should optimise those technological parameters of the process which are possible to change.-Another process improvement can be a certain reconstruction of the tool geometry ensuring lower effort of the die material during forging. We can also consider an additional thermo-chemical treatment, that is the creation of compressive stresses by way of nitriding, which will increase the fatigue strength and may improve the tool’s resistance to the initiation of fatigue cracks.-The observed high cracking tendency of the tool material used prompted the authors to propose a different material for the die (Orvar 2 m produce by Udeholm), which would be more resistant to the propagation of fatigue cracks.-A good alternative in this respect seems to be steel showing higher impact strength, in particular at higher temperatures, as well as the empirically determined value of the stress intensity factor *K*_1C_.-At the same time, increased impact strength may result in greater susceptibility to plastic deformation in the operational surface layer. For this reason, the use of thermo-chemical treatment for it will increase the hardness of its surface, while maintaining a ductile core. The formation of compressive stresses accompanying nitriding should additionally prevent the propagation of fatigue cracks. In total, this should translate into greater durability of the die insert.

## Figures and Tables

**Figure 1 materials-16-04223-f001:**
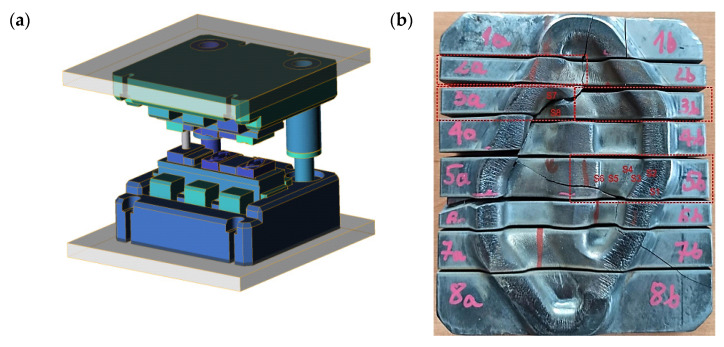
View of (**a**) the CAD model of the tool set and (**b**) a photograph of a worn die, cut into areas subjected to further analyses.

**Figure 2 materials-16-04223-f002:**
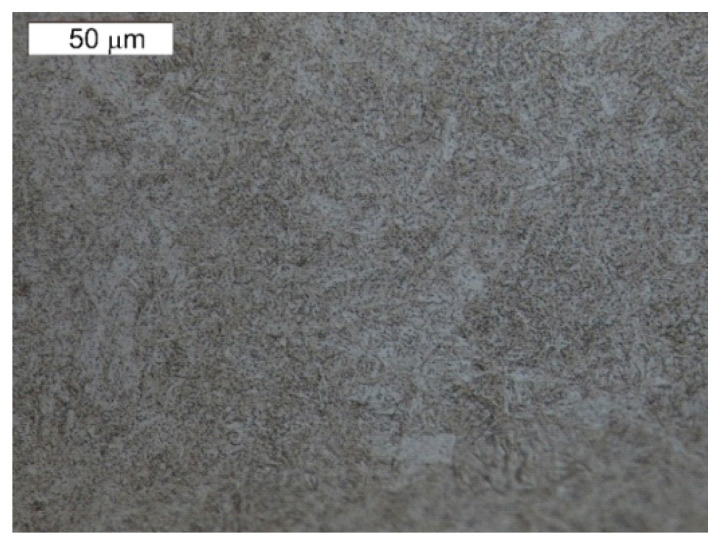
Microstructure of the tested tool’s material. Visible tempering martensite. Light microscopy, etched.

**Figure 3 materials-16-04223-f003:**
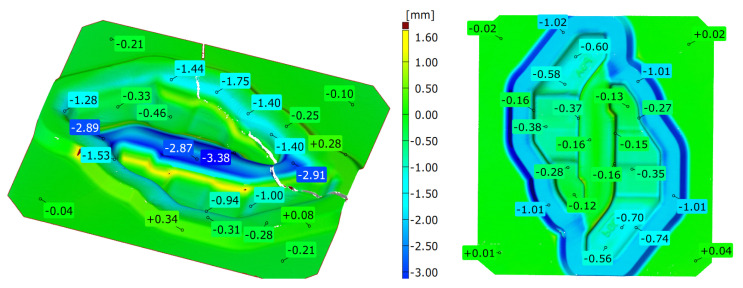
Results of scanning of the working surfaces of the roughing pass.

**Figure 4 materials-16-04223-f004:**
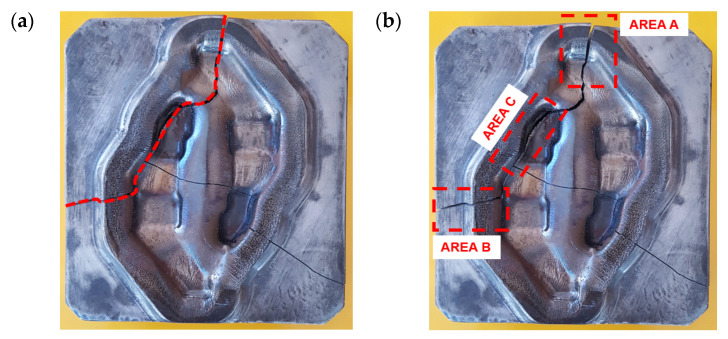
A crack on the die working surface: (**a**) the crack line and (**b**) the selected area in the crack line.

**Figure 5 materials-16-04223-f005:**
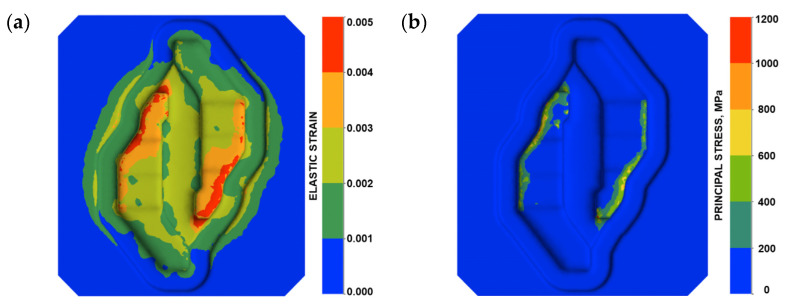
Parameter distribution on the working surface of the tool: (**a**) elastic deformation and (**b**) principal stress.

**Figure 6 materials-16-04223-f006:**
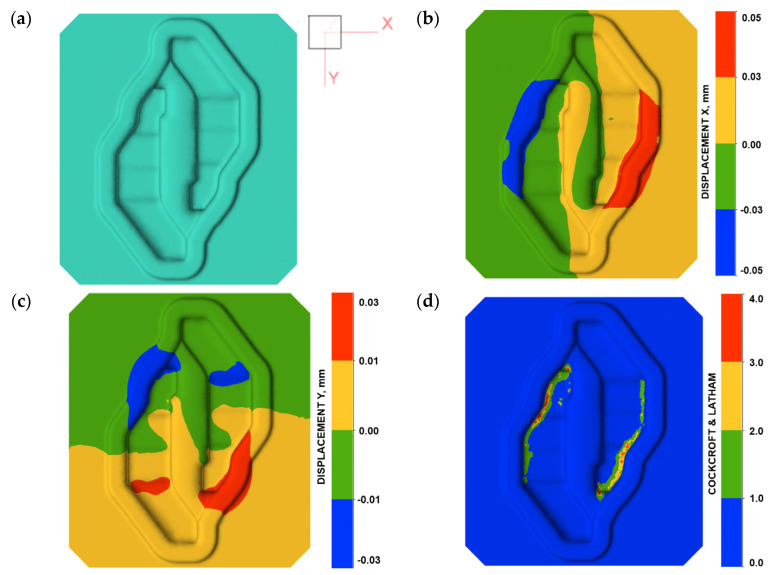
Fracture analysis in Area A and B: (**a**) axis directions, (**b**) displacement X distribution, (**c**) displacement Y distribution, and (**d**) cracking according to the C-L criterion.

**Figure 7 materials-16-04223-f007:**
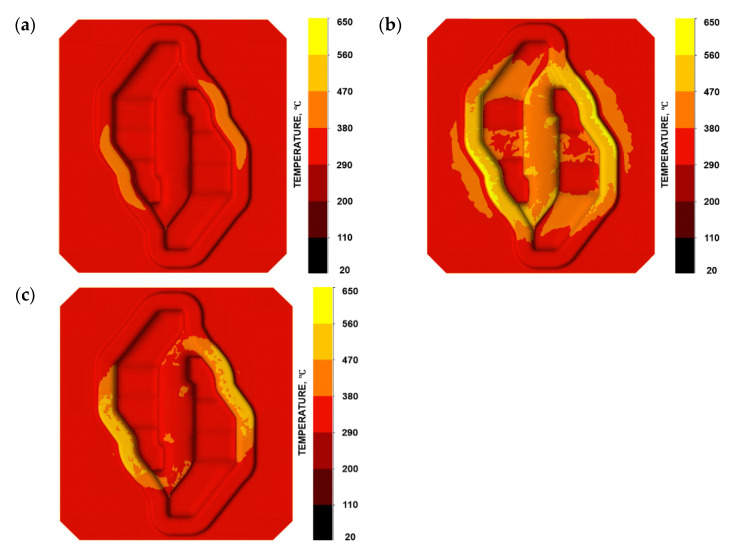
Temperature field distributions (**a**) at the beginning of the forging process—position of the forging, (**b**) initial forming phase, and (**c**) final forming phase.

**Figure 8 materials-16-04223-f008:**
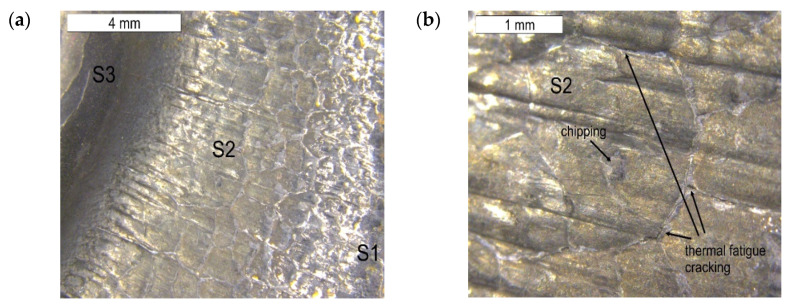
View of (**a**) the operation surface of the tested tool in area 4b. Visible traces of plastic deformation and cracks resulting from thermal fatigue on the external part of the tool; (**b**) a magnified fragment from (**a**).

**Figure 9 materials-16-04223-f009:**
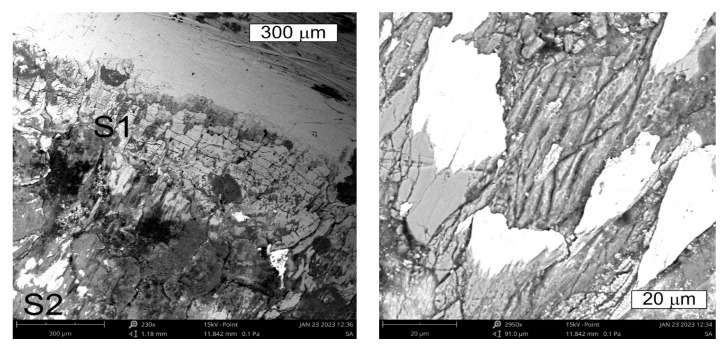
Changes occurring on the tool edge in area S1 of element 5b. Visible spallings and stick-ons on the surface. SEM.

**Figure 10 materials-16-04223-f010:**
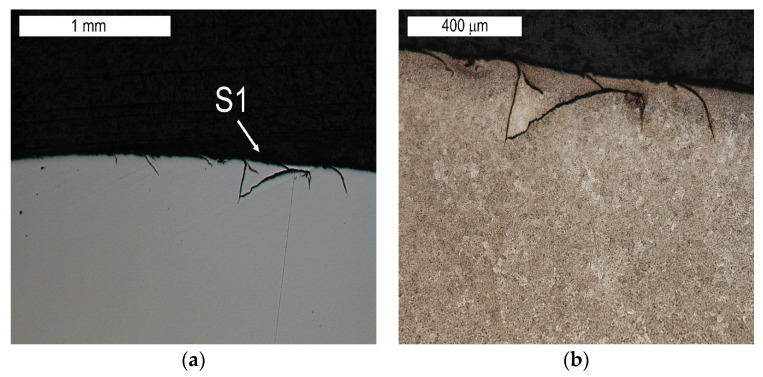
Visible cracks propagating in different directions accompanying the spalling observed in the SEM image in Figure 8 (element 5b). Light microscopy: (**a**) non-etched; (**b**) etched.

**Figure 11 materials-16-04223-f011:**
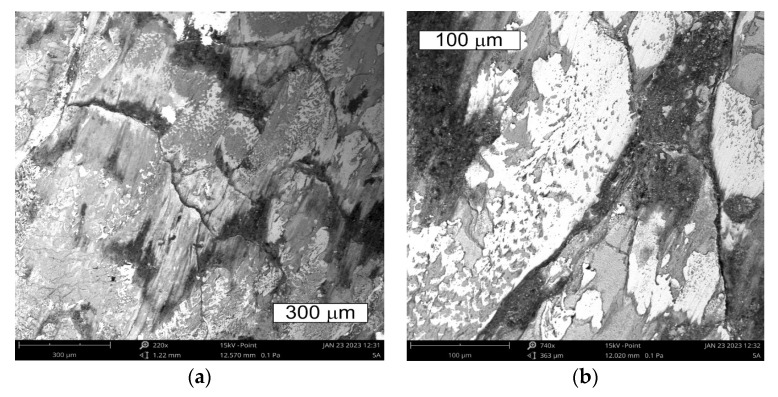
View of (**a**) the thermal fatigue and stick-ons of the forged material on the tool surface in area S2 of element 5b; (**b**) magnified fragments of the area visible in (**a**) SEM.

**Figure 12 materials-16-04223-f012:**
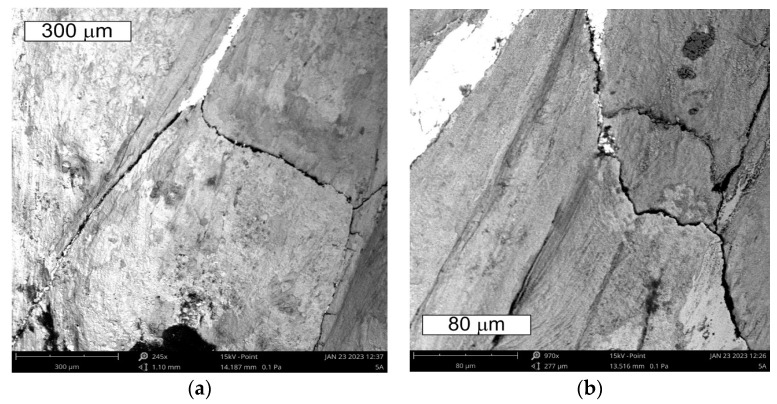
View of (**a**) the thermal fatigue occurring on the tool surface in area S2 of element 5b; (**b**) magnified fragments of the area visible in (**a**) SEM.

**Figure 13 materials-16-04223-f013:**
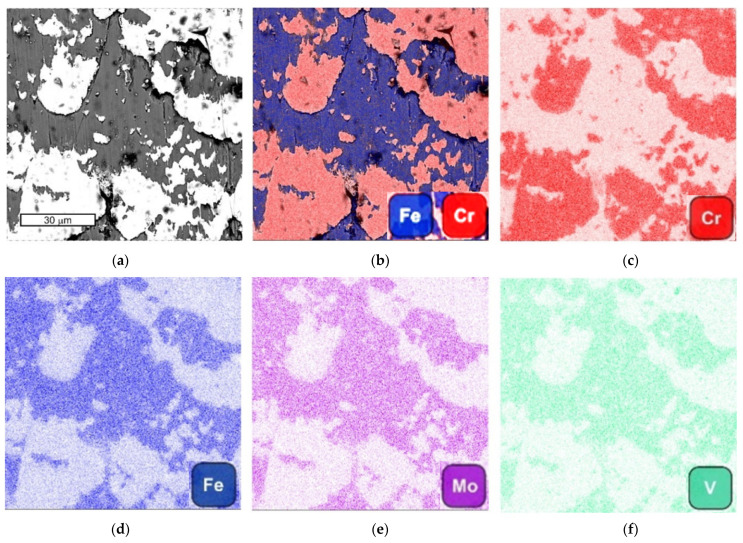
The view of (**a**) SEM image, (**b**) combined SEM image with iron and chromium distribution; (**c**) chromium distribution; (**d**) iron distribution; (**e**) molybdenum distribution; and (**f**) vanadium distribution; SEM/EDX.

**Figure 14 materials-16-04223-f014:**
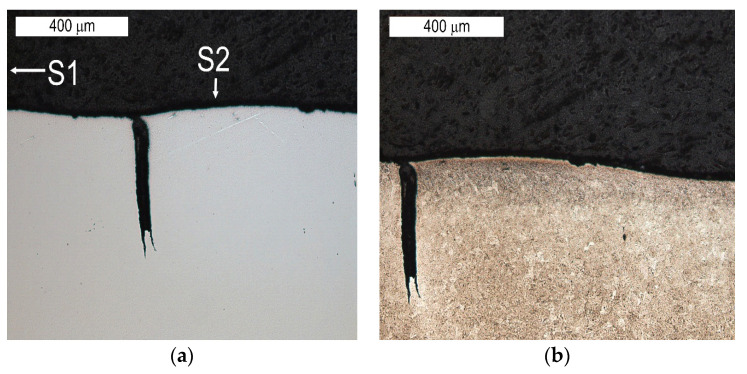
Microstructure of the material in the subsurface area on the upper and external surface of the tools. The cross section made through element 5b. Light microscopy: (**a**) non-etched; (**b**–**d**) etched.

**Figure 15 materials-16-04223-f015:**
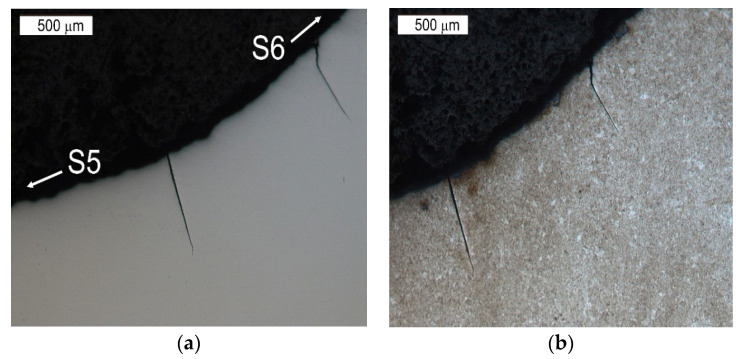
Microstructure in the subsurface area on the transition between the lower and central surface of the used tool. The cross section made through element 5b. Light microscopy: (**a**) non-etched; (**b**) etched.

**Figure 16 materials-16-04223-f016:**
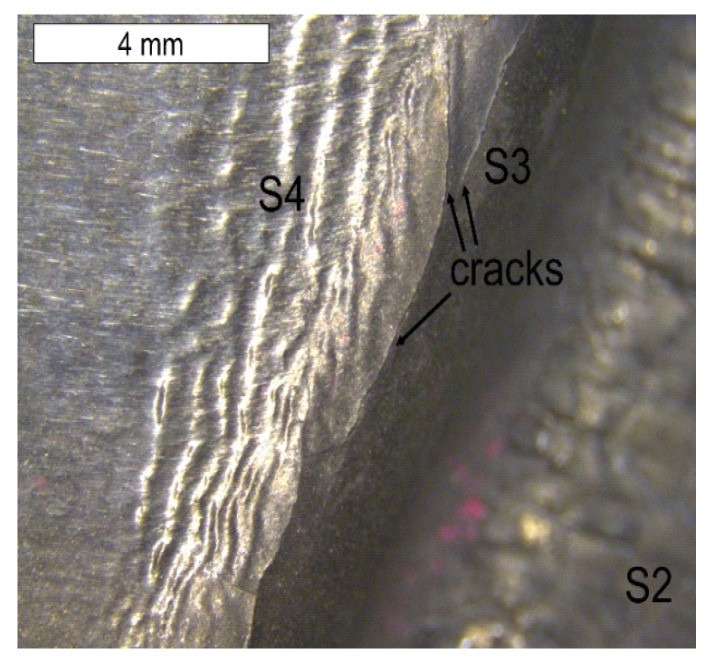
Lower tool surface in the area of element 5b with visible surface wrinkling. Stereoscopic microscopy.

**Figure 17 materials-16-04223-f017:**
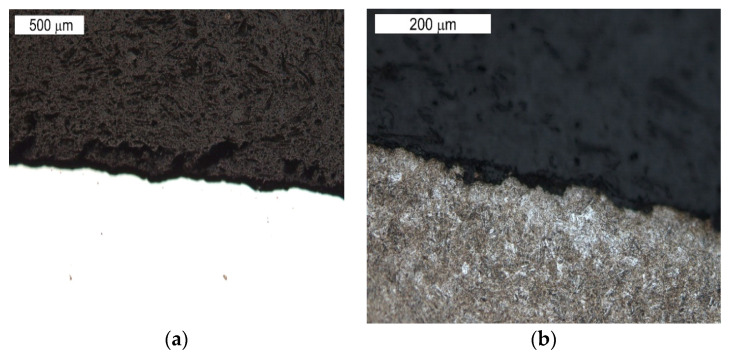
Microstructure of the material in the subsurface area on the lower operation surface of the tool. The cross section made through element 2a. Light microscopy: (**a**) non-etched; (**b**) etched.

**Figure 18 materials-16-04223-f018:**
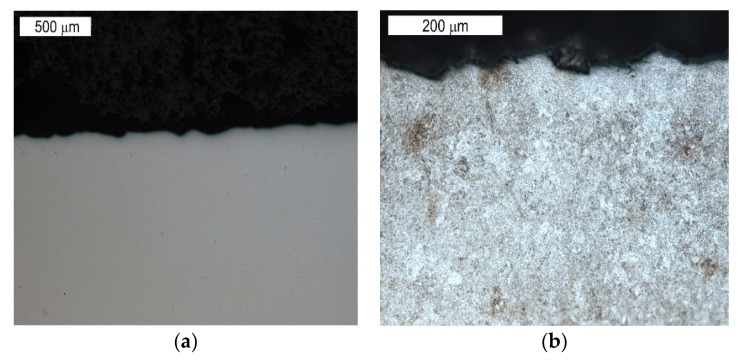
Microstructure of the material in the subsurface area on the central operation surface of the tools (area S6). The cross section made through element 5b. Light microscopy: (**a**) non-etched; (**b**) etched.

**Figure 19 materials-16-04223-f019:**
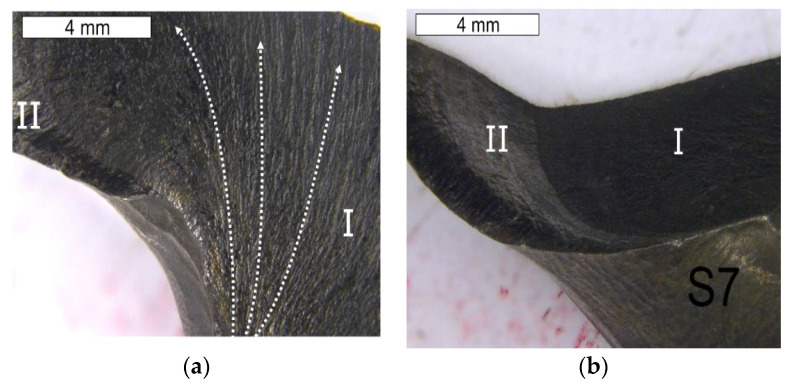
Surface of the fracture of element 3a. Fracture with visible so-called chevron marks, beginning from the area of the crack’s propagation (**a**). A visible auxiliary fault accompanying the fracture shown in Figure a (**b**). Stereoscopic microscopy.

**Figure 20 materials-16-04223-f020:**
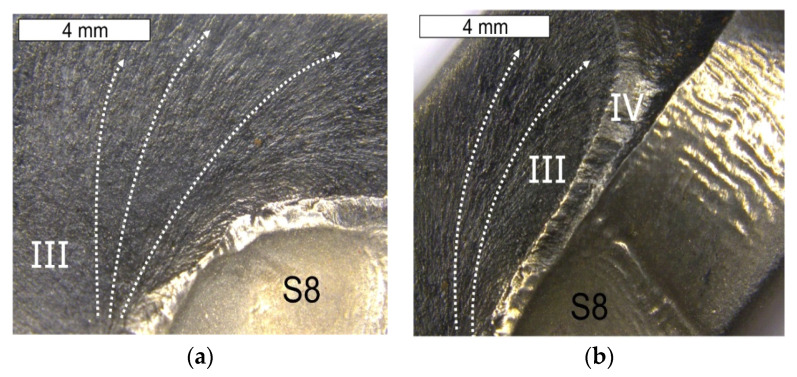
Surface of the fracture of element 3b. Fracture with visible so-called chevron marks, beginning from the area of the crack’s initiation (**a**). Additionally, visible plastic deformations of the surface forming its characteristic wrinkling (**b**). Stereoscopic microscopy.

**Figure 21 materials-16-04223-f021:**
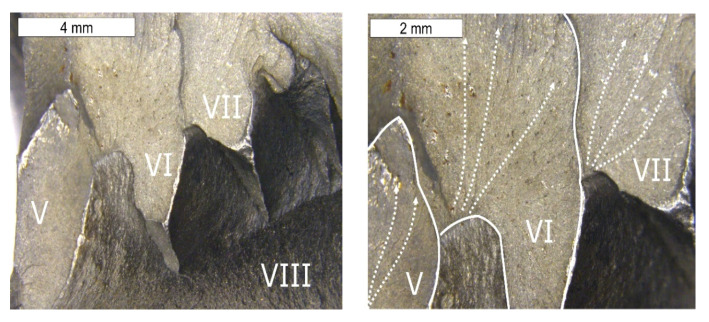
Surface of the fracture of element 2a. A visible multi-plane fracture with clear chevron marks. Stereoscopic microscopy.

**Figure 22 materials-16-04223-f022:**
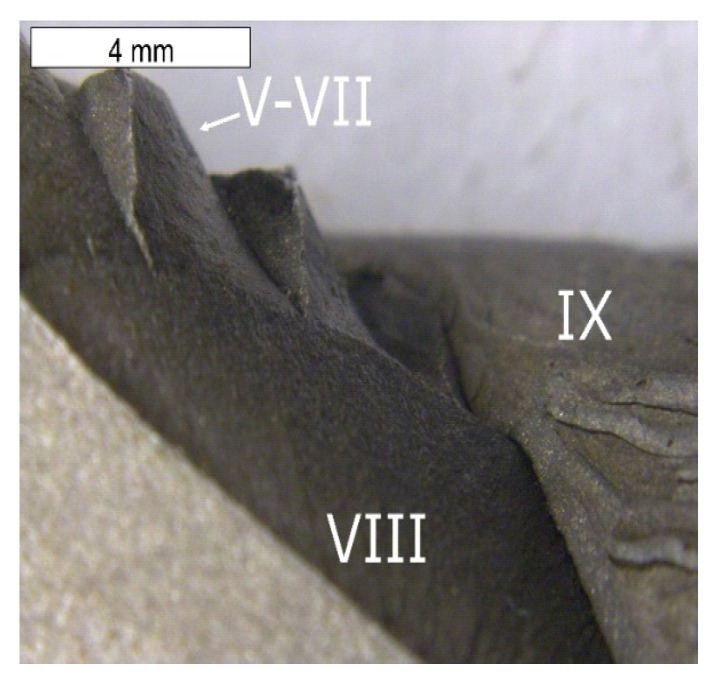
Surface of the fracture of element 2a shown in Figure 20. Visible multi-plane character of the fracture. Stereoscopic microscopy.

**Figure 23 materials-16-04223-f023:**
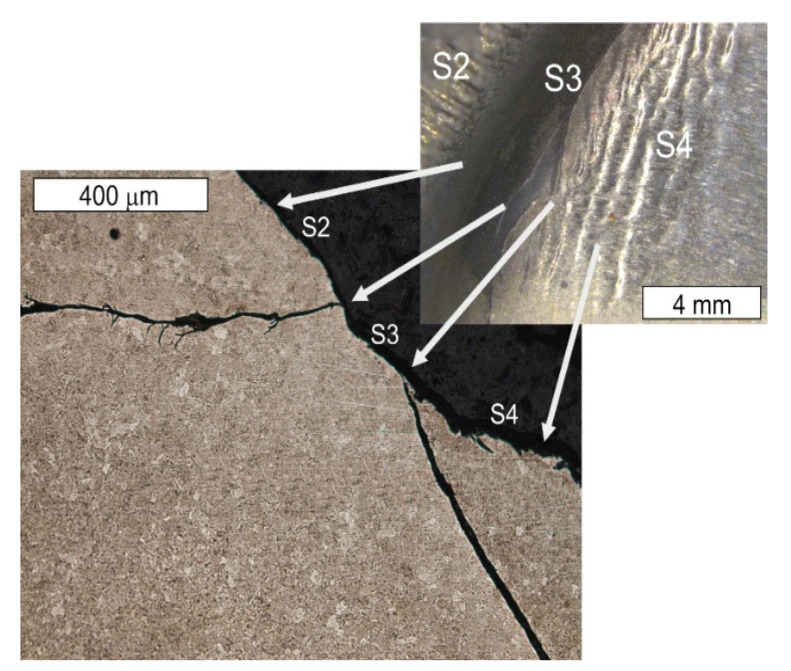
Cracks initiated on the internal tool surface with a macro-image of the area of element 5b, from which a single cross section was made. Light microscopy, non-etched.

**Figure 24 materials-16-04223-f024:**
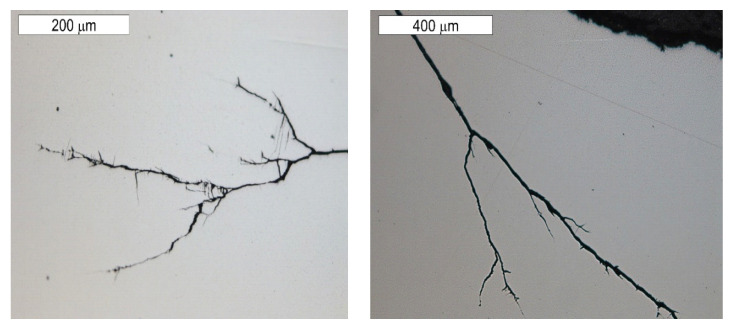
Cracks initiated on the internal tool surface. The cross section made through element 5b. Light microscopy, non-etched.

**Figure 25 materials-16-04223-f025:**
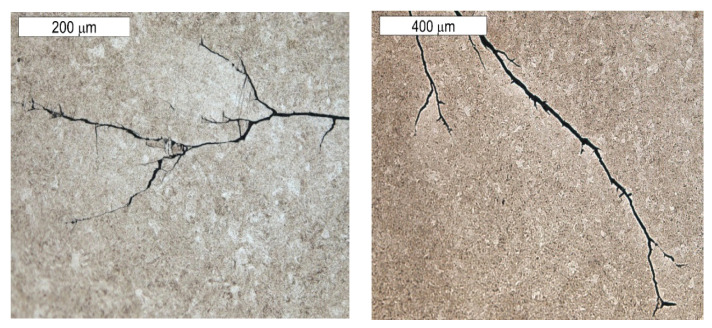
Cracks initiated on the internal tool surface. The cross section made through element 5b. Light microscopy, etched.

**Figure 26 materials-16-04223-f026:**
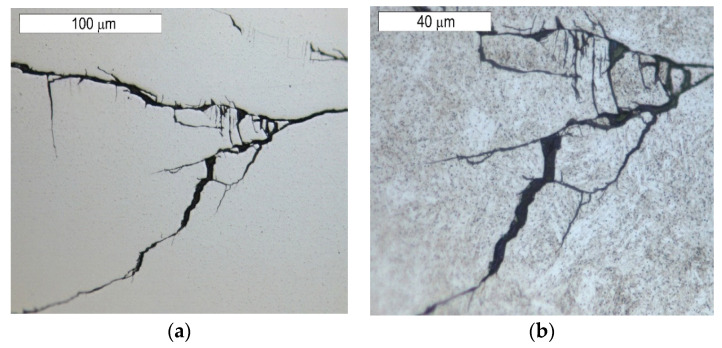
A magnified fragment of the area shown in Figure 18a and Figure 19a. The incubation process of crack formation revealed on the polished surface of the sample. Light microscopy: (**a**) non-etched; (**b**) etched.

**Table 1 materials-16-04223-t001:** Chemical composition of the examined die with similar steel grades.

	C	Mn	Si	S	P	Cr	Cu	Mo	Ni	V	W	Fe
Tested die	0.49	0.45	0.21	0.001	0.006	5.2	0.04	2.33	0.58	0.51	0.01	balance
1.2343	0.33–0.41	0.25–0.50	0.80–1.20	Max. 0.030	Max. 0.030	4.80–5.50	-	1.10–1.50	-	0.30–0.50	-	balance
1.2344	0.35–0.42	0.25–0.50	0.80–1.20	Max. 0.030	Max. 0.030	4.80–5.50	-	1.20–1.50	-	0.85–1.15	-	balance
1.2367	0.35–0.40	0.30–0.50	0.30–0.50	Max. 0.030	Max. 0.030	4.80–5.20	-	2.70–3.20	-	0.40–0.60	-	balance
1.2368	0.38–0.44	0.30–0.50	0.90–1.20	Max. 0.030	Max. 0.030	5.20–5.60	-	2.80–3.10	-	1.10–1.25	-	balance
Unimax	0.50	0.50	0.20	Max. 0.030	Max. 0.030	5.00	-	2.3	-	0.5	-	balance

**Table 2 materials-16-04223-t002:** The results of the breaking work of the V-notch samples and the *K*_1C_ values determined on the basis of impact strength measurements.

Material	No Samples	Temp Test	Energy	*K* _1C_
		°C	J	MPa·m^1/2^
Analysed hot work steel	1	100	15.6	46.6
	2	150	18.2	52.9
	4	200	21.2	60.1
	3	250	22.1	62.2
	5	300	23.3	65.1
Proposed hot work steel	1	100	22.8	63.9
	2	150	28.9	78.5
	3	200	44.3	115.5
	4	250	47.2	122.5
	5	300	53.3	137.1

## Data Availability

Data sharing not applicable.

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
