# Peer review of "Structural Features of Fatigue Crack Propagation of a Forging Die Made of Chromium–Molybdenum–Vanadium Tool Steel on Its Durability"

_materials, 2023, doi:10.3390/ma16124223_

Round 1

Reviewer 1 Report

This work introduces the experimental results of mold inserts used in the pre forging process and proposes an alternative material with higher impact strength. The experimental analysis of this work is comprehensive. I suggest accepting this article after a minor revision.

(1) Please analyze the causes of multi center fatigue fracture. Why multiple centers?

(2) Please provide the surface defect analysis results of the mold insert (such as oxidation, decarburization, etc.).

(3) What is the initial origin of cracks?

(4) Can the new alternative materials meet all usage requirements?

(5) There are some grammar expression issues in the paper, and it is recommended to make revisions.

There are some grammar expression issues in the paper, and it is recommended to make revisions.

Author Response

Dear Editor,

We attach detailed answers in a separate file.

regards,

Reviewer 2 Report

Interesting and original work presented. The work has a clear practical focus. The logic of presenting the results in some places is not clear. The following are the notes:

1. P.1, L. 19: extra dot.

2. P.2, L. 90: It is appropriate to refer to widely used and simple methods for applying protective coatings: arc surfacing method: https://doi.org/10.1016/j.surfcoat.2021.127952 and spark alloying method: https:// doi.org/10.3103/S1068375511040107

3. At the end of the introduction, give the purpose of the work.

4. P. 10, L. 332-333: this statement is not entirely correct. For this statement, it is necessary to present the results of the energy-dispersive analysis of the surface of the stamp, which will indicate the presence of the forged material.

5. Bring the drawings to the same size within each drawing.

6. It is not clear what material the Charpy V-notch impact test was made from. Provide explanations.

7. The Conclusions section needs to be corrected. As presented, this is a short report of the work, not a conclusion.

8. The abstract states that As part of the work carried out, directions for further research were also proposed to improve the durability of the tested tool. It is not clear where they are given in the article. This is partly given in the conclusion, but it should be removed from the conclusion and included in the main content of the article.

Author Response

(The authors gave the same response as above.)

Round 2

Reviewer 2 Report

1. Fig. 13 increase inscriptions

2. Fig. 8, 9, 11, 12 a and b drawings of different sizes, need to be corrected

Author Response

Dear Rewiever,

According to Your suggestion we improved the indicated Figures.

Thanks again for your reviews and comments.

regrads,